# The Onset of Musculoskeletal Pain in the COVID-19 Era: A Survey of Physiotherapy Students in Sicily

**DOI:** 10.3390/jfmk8030091

**Published:** 2023-06-28

**Authors:** Rosario Ferlito, Pierpaolo Panebianco, Valentina Rizzo, Ignazio Prestianni, Marco Sapienza, Martina Ilardo, Maria Musumeci, Vito Pavone, Gianluca Testa

**Affiliations:** 1Department of Biomedical and Biotechnological Sciences, University of Catania, 95123 Catania, Italy; ferlito.rosario@libero.it (R.F.); valentina.rizzo9401@gmail.com (V.R.); 2Department of General Surgery and Medical-Surgical Specialties, A.O.U. Policlinico Rodolico-San Marco, University of Catania, Via Santa Sofia, 78, 95123 Catania, Italy; pierpaolo.panebianco@gmail.com (P.P.); ignazioprestianni93@gmail.com (I.P.); marcosapienza09@yahoo.it (M.S.); martinailardo52@gmail.com (M.I.); musumeci.maria.mm@gmail.com (M.M.); gianpavel@hotmail.com (G.T.)

**Keywords:** musculoskeletal pain, COVID-19 era, physiotherapy studies, study hours, questionnaires

## Abstract

Online teaching has resulted in university students adopting a sedentary lifestyle. Prolonged sitting and reduced physical activity due to pandemic restrictions have led to musculoskeletal pain in various body areas, significantly impacting students’ quality of life. This study aims to investigate the effects of remote learning on Sicilian physiotherapy students during the COVID-19 pandemic, specifically focusing on the occurrence of musculoskeletal pain. An observational study was conducted using an online survey administered through Google Forms. The survey consisted of 26 multiple-choice questions and was distributed to students enrolled in physiotherapy programs at the universities of Catania, Messina, and Palermo. Participants were contacted via social channels or email, and data collection spanned 5 weeks. The collected data were analyzed using R software. A total of 128 questionnaires were collected. At the time of compilation, most respondents (*n* = 103/201, 51.2%) were enrolled in the third year of the course of study in physiotherapy at the universities of Catania, Messina, and Palermo. Their ages ranged between 22 and 25 years (43.3%), and most were female (*n* = 104/201, 51.7%). More than half of the students (51.6%) reported dedicating 15–22 h per week to distance learning for a duration of 6–12 months (50%). Regarding study location, most students preferred studying at a desk (82.8%), and slightly over half (57.8%) adopted a backrest while studying remotely. Analysis of the students’ posture during study hours revealed common positions, including tilting the head forward by more than 20 degrees (47.8%), leaning the trunk forward by more than 20 degrees (71.9%), hunching both shoulders forward (57.0%), wrists positioned above the level of the elbows (46.1%), thighs pointing upwards (41.4%), and one or both feet in a downward or dorsiflexed position (69.5%). In conclusion the questionnaire responses indicate that the lifestyle of university students, influenced by online teaching, has deteriorated, leading to musculoskeletal pain, including myofascial pain. These results are primarily influenced by the adopted posture and the duration of time spent in these positions. Additionally, research is needed to identify the most effective therapeutic approaches for managing musculoskeletal pain.

## 1. Introduction

Throughout history, humanity has encountered numerous pandemics. The challenges associated with managing these pandemics depend on various factors, including their ever-changing and unpredictable nature.

Coronavirus disease 2019 (COVID-19) emerged at the end of 2019, profoundly impacting the global population [1], has resulted in millions of deaths—5,832,333 confirmed deaths on 25 January 2022—and has put a major strain on health systems worldwide [2,3]. The widespread transmission of the virus is evident in its rapid spread, reaching nearly all countries within a span of fewer than 6 months [4].

The illness is characterized mostly by respiratory clinical manifestations ranging from mild to severe clinical forms [5].

Numerous preventive measures have been implemented to mitigate infections, including lockdowns, which have significantly altered our daily routines and have had social, health, and economic implications [6]. Adopting a sedentary lifestyle due to online learning also has implications for musculoskeletal health, as it disrupts our posture [7].

The International Association for the Study of Pain (IASP) defines pain as “an unpleasant sensory and emotional experience associated with actual or potential tissue damage, or described in terms of such damage” [8]. Myofascial pain (MFP) is the leading cause of persistent regional pain, including back pain, shoulder pain, headache, and facial pain [9]. MFP is characterized by localized muscle tension, limited range of motion (ROM), and regional pain [7]. Two studies suggest that MFP is the predominant cause of pain, accounting for 54.6% of chronic head and neck pain [9] and 85% of back pain cases [10].

MFP is characterized by specific clinical features, including the presence of localized, firm, palpable nodules known as trigger points (TrPs) found within taut skeletal muscle bands [11]. Squeezing these TrPs elicits pain, which is typically felt locally or can radiate to distant areas [11]. Imaging techniques, such as X-rays and MRI, do not typically reveal any pathological changes in the muscles or connective tissue [7].

Affected muscles may exhibit stiffness, subjective weakness, pain during movement, and a slightly reduced ROM [9,10,12,13]. Straining the muscles can elicit pain, prompting patients to adopt poor posture and sustain muscle contractions to protect the affected area, contributing to the persistence of pain [14]. Like other chronic pain conditions, MFP is often accompanied by social, behavioral, and psychological disorders that may precede or follow its development [15]. Patients frequently report psychological symptoms such as frustration, anxiety, depression, and anger [7]. Certainly, the transition to distance learning has exposed university students to these conditions [16].

A survey conducted among university students in Rome, Naples, and Bari revealed a significant decline in their physical activity levels during the lockdown period, leading them towards a more sedentary lifestyle [17]. The closure of gyms, sports facilities, and parks as part of social distancing measures made it challenging for students to meet the physical activity guidelines recommended by the World Health Organization [18]. However, it is worth noting that students who were already physically active before the lockdown managed to meet the recommended levels of physical activity [17].

The extended use of technological devices for educational, communication, or entertainment purposes can contribute to prolonged periods of sitting [19]. Suppose the sitting position is maintained for an extended duration, particularly when the trunk is inclined forward or using non-ergonomic chairs [20]. In that case, it can potentially induce or exacerbate lower back pain [19]. Therefore, it is crucial to have an ergonomic workstation to minimize these risks [20].

Maintaining a correct and relaxed posture involves several key elements. It includes keeping a straight gaze and slightly tilting the head forward, with the trunk positioned upright or slightly reclined and adequately supported by the backrest [21]. The shoulders should not be raised unless the countertop is too high [21]. The arms and the elbows resting on the table should form an angle of at least 90° with the forearms, while the thighs should be parallel to the ground or slightly inclined forward under the work surface [21]. The angle between the thighs and lower legs should be approximately 90°, allowing for ample movement of the thighs and knees in an upward, forward, and sideways direction [21].

Moreover, there should be enough room to comfortably stretch the legs, and the chair’s edge should not compress the thigh muscles or the area behind the knees [21]. The feet should rest flat on the floor or a footrest, and they should be free to move forward, sideways, and backward [21]. This can be achieved using an ergonomic workstation [22]. There is also a biopsychosocial model in which the focus is on promoting a healthy lifestyle that includes regular physical activity, stress reduction exercises, a balanced diet based on the Mediterranean diet, and sufficient restful sleep [7,8,12,15].

Lack of physical activity is associated with decreased cardiorespiratory and muscular functions, including muscle strength and mass [15].

For individuals engaged in sedentary work and spending prolonged periods in a seated position, it is recommended to incorporate regular breaks with a 2 min walk every 20–30 min [7,8,12,15].

This study aimed to examine the experiences of university students during this significant historical period, which will undoubtedly leave a lasting impression in the collective memory.

The novelty of the study is to try to identify the correlations between people’s postures during the pandemic and the associated musculoskeletal disorders.

## 2. Materials and Methods

### 2.1. Design of the Study

The present work is an observational study conducted through an online survey using the “Google Forms” platform (USA). The questionnaire comprises 26 multiple-choice questions. The first four questions (Q1–Q4) aim to gather demographic information from the participants, providing insights into general characteristics of the target population, such as the year of their degree course, age, gender, and lifestyle. (Table 1) The subsequent 20 questions (Q5–Q24) delve into specific aspects of online teaching activity, including time spent distance learning and studying, tools and environments utilized for studying, the posture adopted, and musculoskeletal pains and disorders experienced. These questions aim to understand students’ approach to this novel learning mode and its potential consequences. Finally, the last two questions (Q25–Q26) explore the psychosocial aspect, evaluating students’ self-assessment of their quality of life and overall experience during this period.

### 2.2. Participants and Setting

The survey was administered to a regional sample of students who are currently enrolled in physiotherapy programs at the universities of Catania, Messina, and Palermo. Participants were recruited through various social channels and email. The inclusion criteria for the study were as follows: proficiency in the Italian language and regular enrollment in the physiotherapy program at the universities of Catania, Messina, and Palermo.

### 2.3. Collection Procedure

The online survey was conducted using the “Google Forms” platform (USA) at www.google.com (accessed on 25 June 2023). Participants were contacted through social channels, specifically WhatsApp or email. Those reached through social channels received the questionnaire link directly, while others were sent an email containing the survey link. In both cases, participants were provided with a brief explanatory note outlining (1) the study’s purpose, (2) data management procedures, including anonymization, (3) an informed consent statement, and (4) an invitation to complete the survey. Participation was voluntary, and no incentives were offered to participants. They were instructed to respond to all questions to ensure complete data. Participants could revise or change their answers before finalizing the questionnaire. The researchers maintained the anonymity of participants, ensuring confidentiality and data protection to prevent any psychological harm. The collected data were subsequently forwarded to a statistician (G.R.) who analyzed the data. Data collection was carried out over a period of 5 weeks, starting on 30 August 2021, and concluding on 4 October 2021.

### 2.4. Statistical Analysis

The raw data from Google Forms were extracted and subsequently exported to Excel for statistical analysis. A descriptive analysis was performed, calculating frequencies and relative percentages of the various variables relevant to the survey. The aim was to assess whether there were differences among subjects based on the duration of their engagement in distance learning and the intensity of pain reported on the Visual Analog Scale (VAS). Specifically, a one-way analysis of variance (ANOVA) was utilized, which is a statistical method suitable for testing differences between the means of three or more groups. In our case, the three groups were defined based on the duration of distance learning: 1–6 months, 6–12 months, and over 12 months. One of the assumptions underlying the ANOVA method is the homoscedasticity or equal variance between groups. To test this hypothesis, Leneve’s test was employed as an alternative to Bartlett’s test, as it is less sensitive to deviations from normality. All data were analyzed using R Software (Version 4.1.1).

## 3. Results

A total of 201 questionnaires were collected. At the time of compilation, most respondents (*n* = 103/201, 51.2%) were enrolled in the third year of the course of study in physiotherapy at the universities of Catania, Messina, and Palermo. Their ages ranged between 22 and 25 years (43.3%), and most were female (*n* = 104/201, 51.7%).

More than half of the students perceived themselves as having low levels of sedentary behavior. Specifically, 18.9 percent considered themselves not at all sedentary, 56.7 percent described themselves as a little sedentary, 23.4 percent considered themselves quite a bit sedentary, and only 0.5 percent indicated being very or very much sedentary.

### 3.1. Time Spent Distance Learning/Studying (Questions 5–7)

The majority of respondents (57.7%) reported spending 15–22 h per week distance learning, followed by 7–14 h for 27.4%, 23–30 h for 10.4%, and a small minority (4.5%) who spent more than 30 h (Figure 1). These figures are related to the duration of distance learning; 48.8% of respondents had started 6–12 months ago, 42.3% had started more than 12 months ago, and only 9.0% had devoted 1–6 months to it. Additionally, examining the number of hours students dedicate to self-studying independently from classes is important. For most college students in our surveys (59.7%), this amounts to 4–6 h per day, while 19.9% dedicate 1–3 h, and 19.4% allocate 7–10 h. A very small percentage of respondents (1.0%) reported spending more than 10 h on self-study.

### 3.2. Study Tools Used (Questions 8–10)

Considering that electronic devices have become integral to our daily lives, it is important to assess the amount of time spent using them. A majority of respondents (58.2%) reported spending 1–3 h on electronic devices, followed by approximately one third (34.3%) who spend 4–6 h, and a small percentage (7.0%) who spend 7–10 h. Only 0.5% of respondents reported spending more than 10 h on electronic devices. Interestingly, despite the prevalence of digital technology, the classic book remains a popular study tool, utilized by almost everyone (97.0%), closely followed by personal computers (PCs) (96.0%). PCs are predominantly used without any additional support (78.1%). Additionally, cell phones are relied upon by 50.2% of respondents, while tablets are used the least (16.9%) (Figure 2).

### 3.3. Environment and Posture Adopted during Study (Questions 11–20)

When it comes to study environment, the majority of students prefer to study at a desk (85.1%), while only a small percentage indicated studying on a bed in a sitting position (3.0%), semi-sitting (7.0%), lying down (2.5%), or on a sofa (2.5%). During distance learning or study hours, slightly more than half of the students (69.7%) tend to rest their back on the backrest, maintaining an optimal distance of approximately 50–70 cm from the monitor (67.2%). However, 23.4% maintain a shorter distance than recommended, while only 9.5% maintain a greater distance. Different body segments assume various positions, with most respondents tilting their heads forward by more than 20° (53.2%) and leaning forward (31.8%). Only a small percentage tilt their heads to the side (7.0%), recline backwards (5.5%), or turn their heads to one side by more than 20° (2.5%).

The majority of subjects (77.1%) maintain a forward trunk tilt of more than 20°, while a portion reclines it backwards by more than 20° (10.4%) or bends it to one side (10.0%). A small percentage of subjects turn their trunk to one side by more than 20° (2.5%). In terms of shoulders, a significant proportion (61.7%) frequently have one or both shoulders leaning forward, while the remaining 38.3% tend to have raised shoulders.

Regarding the position of the wrists, the responses show significant variation: 52.2% of respondents keep both wrists positioned above the level of their elbows, while 17.4% have both arms forming an angle of more than 20° concerning the trunk. Furthermore, 19.4% maintain only one wrist above the elbow level, and for 10.9%, one arm forms an angle of more than 20° concerning the trunk (Figure 3).

### 3.4. Pain Associated with Distance Learning (Questions 21–24)

Many students report experiencing pain primarily in the lumbar region (73.1%) and cervical area (69.2%), and headaches (60.7%) are also common. Shoulder pain (28.4%), back pain (27.9%), wrist pain (8.5%), and elbow pain (4.5%) are relatively less prevalent (Figure 4).

Almost all participants reported experiencing stinging pain (61.2%), often accompanied by numbness (59.2%). A smaller percentage of participants mention tingling (18.4%) or burning (15.9%) sensations, while an even smaller proportion report stabbing pain (8.5%) (Figure 5).

When university students were asked to rate the intensity of their pain on a VAS ranging from 0 to 10, the average score obtained was 4.4, with a median of 5. The reported scores ranged from a minimum of 0 to a maximum of 9 (Figure 6).

Slightly more than half of the participants (58.7%) reported experiencing pain only after sitting for an hour. About 17.9% reported pain after 30 min of sitting, 15.4% after 20 min, 7.0% after 10 min, and 1% after 5 min (Figure 7).

### 3.5. Distance Learning Experience (Questions 25–26)

When asked to share their opinion on the experience of distance learning during the COVID-19 pandemic era, over half of the interviewed sample (61.7%) stated that their quality of life had worsened compared to the period when they attended in-person classes. Only 5.5% of students considered their quality of life to have improved, while 32.8% did not notice any significant difference between the two learning modes. For most university students, this period will be remembered as a negative experience (64.7%), while a smaller portion viewed it positively (35.3%).

No significant difference was observed between subjects based on the duration of distance learning in relation to pain intensity scores measured using the VAS scale. One-way ANOVA was employed, considering three groups based on the duration of distance learning: 1–6 months, 6–12 months, and >12 months. The hypotheses for the ANOVA test are as follows. Null hypothesis: The means of the different groups are equal.

Alternative hypothesis: At least one sample mean is different from the others.

Table 2 presents the mean and standard deviation (SD) of the VAS scale scores for each of the three groups based on the duration of distance learning (Figure 8 and Table 2).

The objective is to determine whether there is a significant difference between the mean scores of the VAS scale among the three groups based on the duration of distance learning. The ANOVA model obtained is the following (Table 3).

The *p*-value is 0.18. It can be asserted that since the *p*-value is greater than 0.05, there are no significant differences between the three groups.

## 4. Discussion

More than half of the students (51.6%) reported dedicating 15–22 h per week to distance learning for a duration of 6–12 months (50%). The requirement to attend online lessons necessitated spending extended periods in front of computers, tablets, and mobile phones, leading to a sedentary lifestyle that is considered inappropriate for their well-being. Regarding study location, most students preferred studying at a desk (82.8%), with slightly over half (57.8%) adopting a backrest while studying remotely. Additionally, the hours dedicated to distance learning should be taken into account alongside the time spent on studying, which averaged around 4 to 6 h for the majority of students, as well as the hours spent on electronic devices for entertainment purposes, which ranged from 1 to 3 h for over half of the university students. This aspect is significant due to the widespread internet addiction observed globally. According to the Global Digital Report 2021, Italy’s daily online usage in 2021 was slightly below the world average, with individuals spending 6 h and 22 min online, which is 22 min more than the previous year. Naturally, the COVID-19 pandemic has influenced the variation in these statistics [23].

To support this thesis, it is evident that the preferred study tools among the participants are books and PCs, despite most students not utilizing PC support. Additionally, almost all participants typically study at a desk, leaning back, which facilitates adopting a posture that is as close to neutral as possible rather than studying on a sofa or bed. However, an article by Heidi Mitchell published in the “Wall Street Journal” suggests that it is possible to study in bed while adhering to certain guidelines.

Among the experts in the field, Atul Malhotra, a physician at the University of California, has reported that lying down or sitting does not alter brain activity, indicating that posture does not affect brain functioning [24]. Janice Fletcher, an ergonomics expert at a medical center in San Diego, has also provided insights for this research. Fletcher explained that sitting up in bed with pillows supporting the arms can reduce the effort required to hold reading material, making it an ideal position. Furthermore, Fletcher advised individuals to prioritize comfort, as uncomfortable individuals are more prone to distractions [24].

More than half of the students in the study maintain a distance of 50–70 cm from the monitor, as recommended by guidelines. They also study in a well-lit environment that ensures appropriate contrast between the screen and the room. The lighting should facilitate easy reading of the screen and clear recognition of keyboard characters. However, it is important to avoid excessively bright light that may make it difficult to read the information displayed. If fluorescent lamps are utilized, it is advisable to opt for neutral-white or warm-white ones, as they provide greater comfort and create a pleasant ambiance, making the environment cozier [25].

When considering different parts of the body, it was observed that most interviewees tend to tilt their head and trunk forward by more than 20°. However, a correct and relaxed posture necessitates a slight forward tilt of the head while keeping the trunk straight or slightly reclined with proper backrest support. Slightly over half of the students tend to tilt their shoulders forward instead of retracting them. This posture is commonly adopted when limited space restricts leg movement, when the work surface is positioned too far, or when working on a computer with documents placed on the keyboard. Respondents typically maintain their wrists positioned above the level of the elbows, which reduces strain on the forearm muscles that insert at the elbow and lowers the risk of developing carpal tunnel syndrome. The angle formed by the thigh and leg should ideally be 90°, with the feet resting flat on the floor or on footrests to achieve a slight dorsiflexion. However, most students tend to point their thighs downward due to higher seating positions while still maintaining their feet in an optimal position [21].

As stated by J. Fricton, poor posture, lack of physical activity, anxiety, and tension are all risk factors for musculoskeletal pain [7].

The most common symptoms reported by the interviewees are pain in the lumbar and cervical spine, followed by headaches. Chronic neck pain is recognized as a public health concern, and a significant proportion of muscle pain is associated with myofascial pain syndrome, characterized by the presence of myofascial TrPs [19]. In addition, S. T. Celenay, Y. Karaaslan et al. note that prolonged periods of sitting can also contribute or aggravate lower back pain by increasing pressure on the intervertebral discs and causing nutritional imbalances (Figure 9, Figure 10 and Figure 11) [19].

While genetics plays a significant role in the predisposition to headaches, various factors can trigger them, including emotional and physical stress, particularly during pandemic-related restrictions, the sedentary lifestyle associated with distance learning, and eye fatigue resulting from prolonged use of electronic devices [26].

The pain experienced is primarily described as a stinging sensation or a feeling of numbness. Myofascial pain is characterized by the presence of active or latent TrPs. Active TrPs refer to points that cause spontaneous pain without the need for palpation [11]. Typically, these TrPs are not associated with muscle weakness or sensory deficits unless there is the involvement of the nervous system caused by nerve root compression [9,26].

Pain is an inherently personal and subjective experience, making assessing all its aspects through objective measurements challenging. To address this, we employed the VAS, a quick and simple method that allows subjects to visually represent their pain level. It can be completed in less than a minute. Many university students reported an average pain score of 4.4, primarily experienced after sitting for an hour. This response aligns with the existing literature, as a seated position places the spine under the weight of the head, trunk, arms, hands, and any objects held. Consequently, the ligaments in the spinal column are subjected to shear and compressive forces.

Applying compressive forces to various parts of the body hampers the exchange of metabolic products and nutritional inputs with the circulatory system, accumulating lactic acid, fatigue, and pain [13]. Specifically, the neck and back muscles remain active during the seated position. Prolonged static contraction of these submaximal muscles can lead to muscle tension and fatigue. Fatigue in the paraspinal muscles diminishes their ability to support the spine, increasing mechanical stress on the ligaments and intervertebral discs and affecting motor coordination and muscle control. However, P. Waongenngarm et al. reveal generally conflicting evidence in this regard [13].

There is currently no evidence to suggest a difference in pain intensity scores on the VAS scale among subjects based on the duration of months dedicated to distance learning. This lack of differentiation may be attributed to the relatively short duration of the distance learning period. Therefore, it would be valuable to conduct future studies that involve tracking students over a longer timeframe to gain deeper insights into this relationship.

More than half of the participants reported that their quality of life has deteriorated compared to when they were attending classes, leading them to consider this period a negative experience (Figure 12).

This can be attributed to the negative impact of global tragedies and diseases, such as COVID-19, on mental health. These circumstances often give rise to symptoms of stress, depression, anxiety, and post-traumatic stress. As a result, individuals may develop self-protective mechanisms, including feelings of fear and other negative emotions.

Humans generally feel comfortable whenever they are able to recognize, explain, predict, and control any ongoing process. If this is not possible, this increases uncertainty, confusion, dysfunctional behavior, and fragility, as reported by M. Jakovljevic et al. [27].

Indeed, distance learning has brought about a radical revolution in the lives of students and their university experience, as the internet simultaneously brings people closer and separates them [6]. Consequently, the personal interactions, physical presence, and connection with the social and urban environment surrounding university have been lost. As a result, there is a risk that students may easily lose interest in studying. However, it is important to note that distance learning also has its advantages. For instance, students no longer need to navigate through city traffic or rent accommodation if they are non-residents, resulting in economic savings. Additionally, the convenience and ease of attending classes remotely have made the period of distance learning a positive experience for some students.

Distance learning was primarily embraced by first-year students in the study course, likely because they commenced their university journey during the COVID-19 restrictions, thus lacking a point of comparison between university life before and after the pandemic. Conversely, nearly all third-year students held a different perspective (Figure 13).

To the best of our knowledge, this study is the first of its kind in the literature, examining the impact of remote work on Sicilian physiotherapy students during the COVID-19 pandemic era. It gathers data on both the musculoskeletal aspects and the subjective experiences of the students. Furthermore, this study serves as a foundational point for future research exploring the effects of online education on university students worldwide.

The study had several limitations. It included only Sicilian university students who were predominantly recruited through the “WhatsApp” application. The exact percentage of active participants on this social network remains unclear. Additionally, there is a possibility of recall bias and social desirability bias due to the self-reported and retrospective nature of the data.

## 5. Conclusions

The data collected in this study indicate that most university students perceive the period of distance teaching as a negative experience, aligning with the observed decline in their overall quality of life.

While the majority of participants maintain an optimal head inclination and maintain a recommended distance of 50–70 cm from the monitor in a well-lit environment, their posture concerning the wrists, feet, shoulders, and thighs does not adhere to the guidelines for proper posture in sedentary work. This finding is further compounded by the lack of utilization of ergonomic devices to establish an ergonomic workstation among university students.

The combination of these factors leads to the generation of pain, primarily affecting the cervical region and lower back and resulting in headaches, with the most prevalent form being myofascial pain.

The novelty of the study was to try to identify the correlations between the postures taken during the pandemic and the associated musculoskeletal disorders.

Further clinical studies are required to investigate the risks posed to both mental and physical health due to a compelled sedentary lifestyle. Additionally, research is needed to identify the most effective therapeutic approaches for managing musculoskeletal pain.

## Figures and Tables

**Figure 1 jfmk-08-00091-f001:**
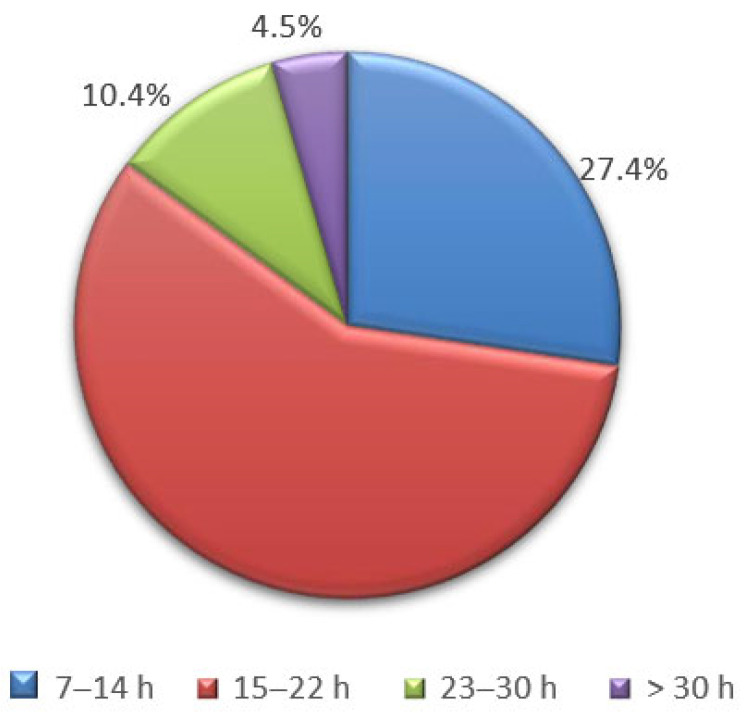
Graph of Q.5: Weekly hours of distance learning.

**Figure 2 jfmk-08-00091-f002:**
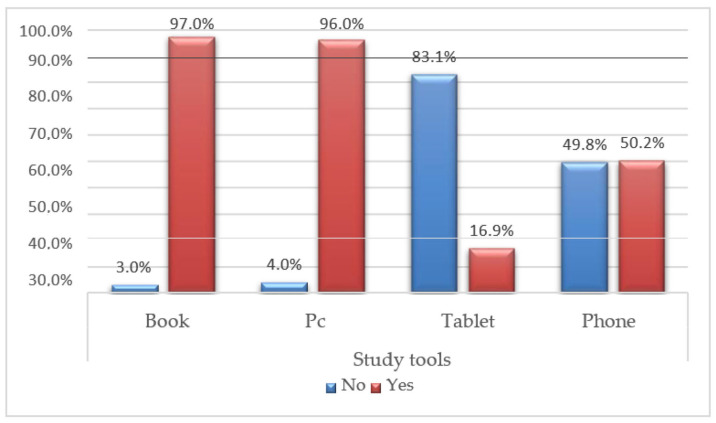
Bar graph of Q.9: Study tools.

**Figure 3 jfmk-08-00091-f003:**
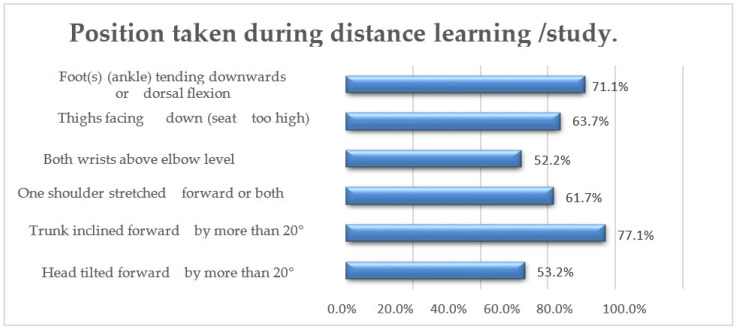
Bar graph of the most assumed positions during distance learning/study hours.

**Figure 4 jfmk-08-00091-f004:**
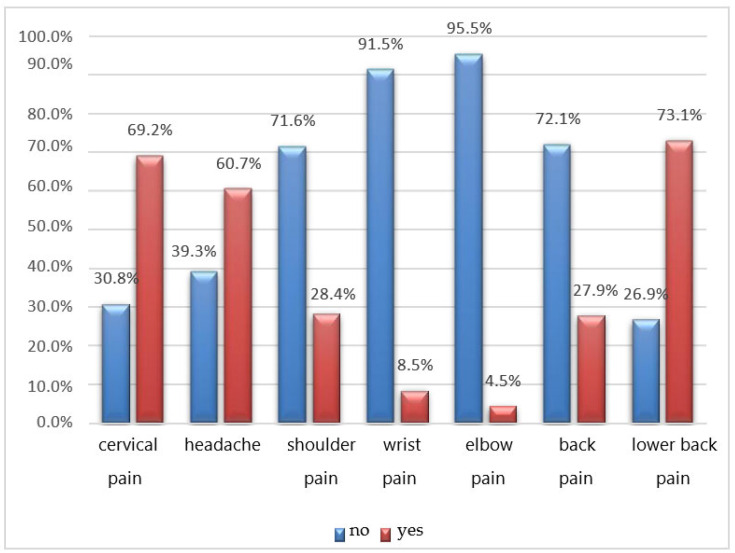
Q.21 bar graph: If you report disturbances, where?

**Figure 5 jfmk-08-00091-f005:**
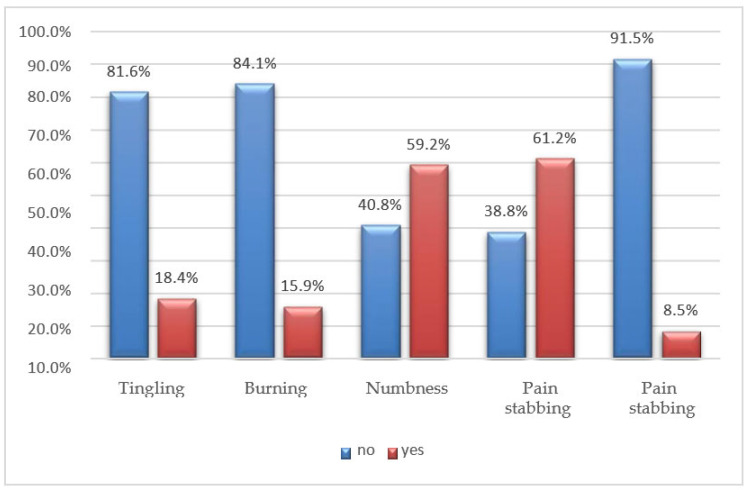
Q.22 bar graph: What kind of pain do you feel?

**Figure 6 jfmk-08-00091-f006:**
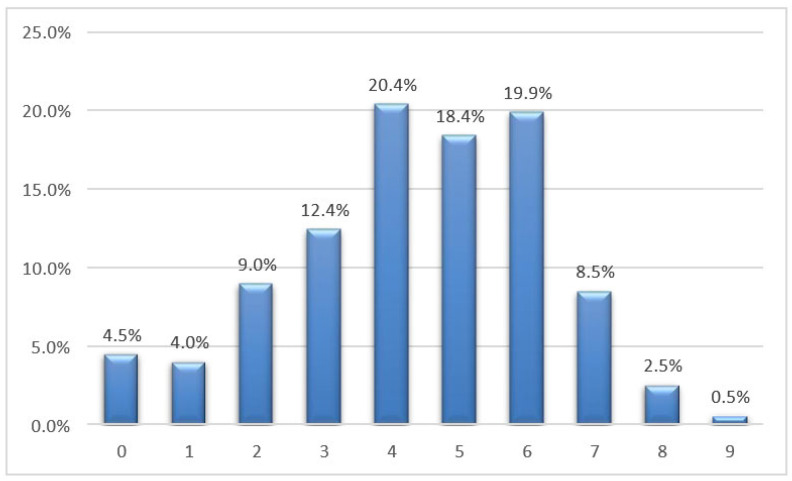
Bar graph of Q.23: Express with a score from 0 to 10 the intensity of pain—VAS scale.

**Figure 7 jfmk-08-00091-f007:**
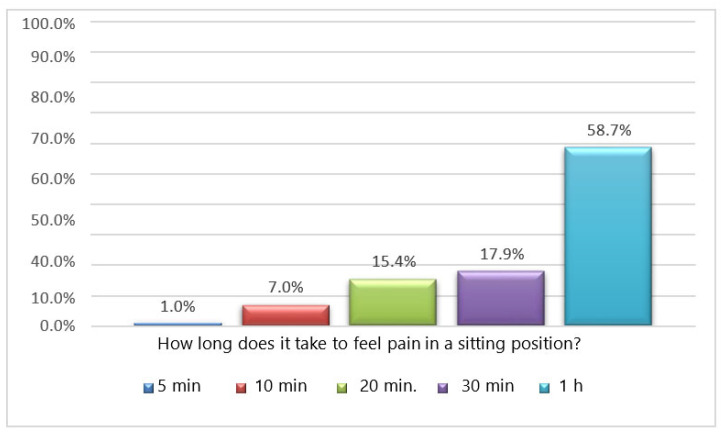
Q.24 bar graph: How long does it take you to feel pain in a sitting position?

**Figure 8 jfmk-08-00091-f008:**
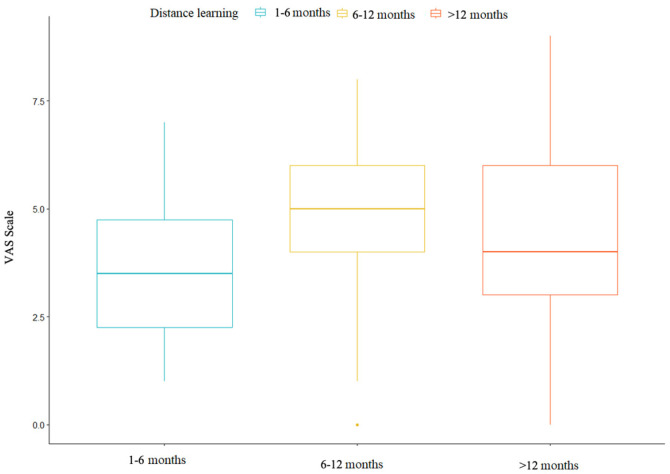
Three box plots relating to the months of distance learning.

**Figure 9 jfmk-08-00091-f009:**
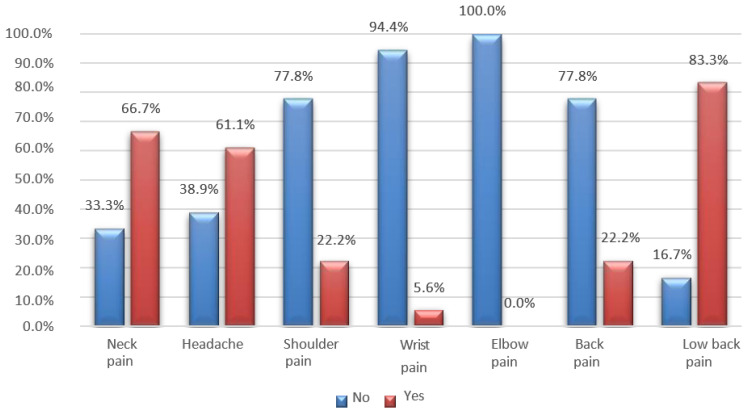
Bar graph of pain reported by students who spent 1–6 months in distance learning.

**Figure 10 jfmk-08-00091-f010:**
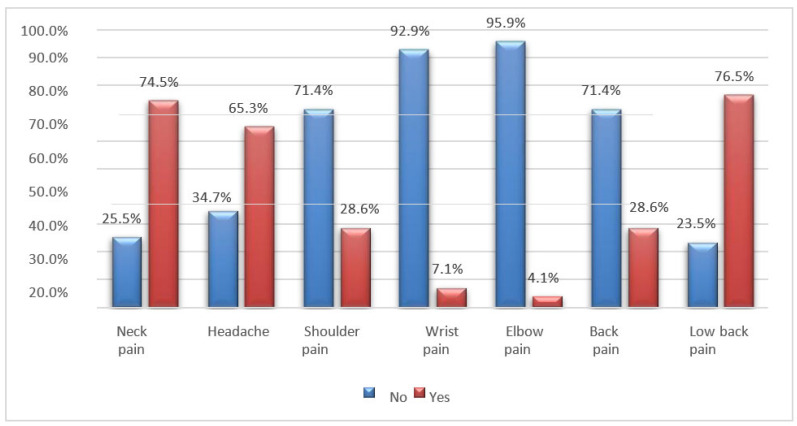
Bar graph of pain reported by students who spent 6–12 months in distance learning.

**Figure 11 jfmk-08-00091-f011:**
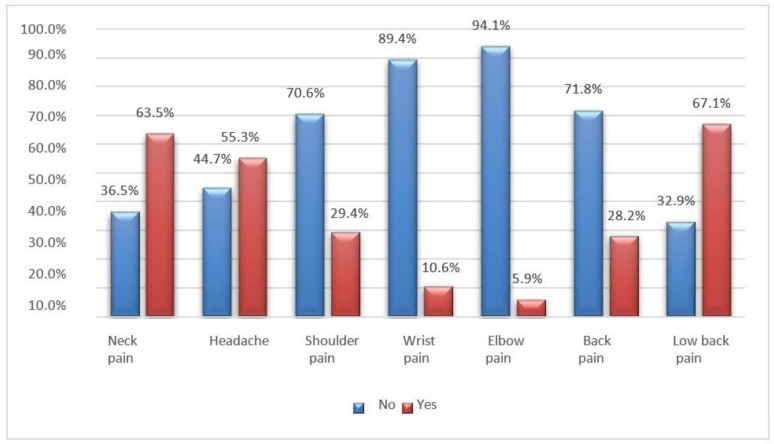
Bar graph of pain reported by students who spent more than 12 months in distance learning.

**Figure 12 jfmk-08-00091-f012:**
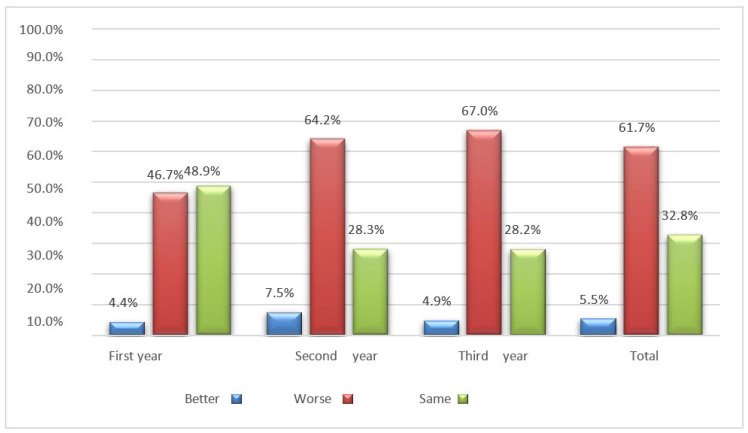
Bar graph of students’ quality of life according to the year of the study program.

**Figure 13 jfmk-08-00091-f013:**
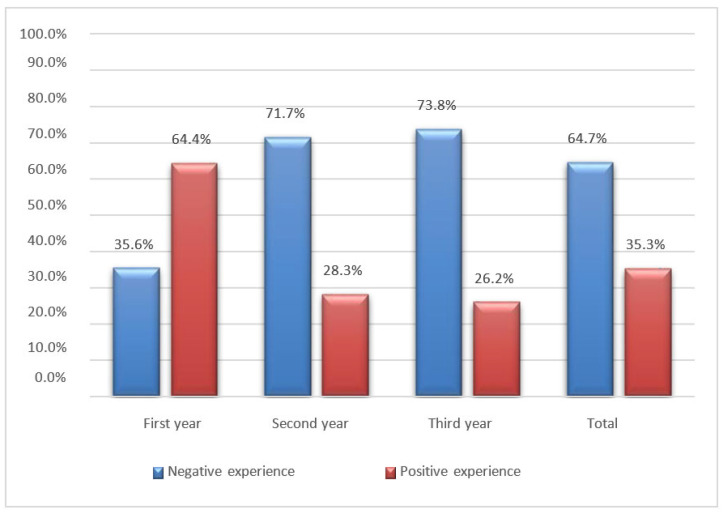
Bar graph of the students’ distance learning experience according to the year of the study program.

**Table 1 jfmk-08-00091-t001:** Results Q1–Q4.

Variables	Total Number of Respondents (*n*=)	%
**Year of the Degree Course**		
First	45	22.4
Second	53	26.4
Third	103	51.2
**Age**		
18–21 years	84	41.8
22–25 years	87	43.3
26–29 years	23	11.4
>30 years	7	3.5
**Gender**		
Man	97	48.3
Woman	104	51.7
**You consider yourself a sedentary person**		
Not at all	38	18.9
Little	114	56.7
Enough	47	23.4
Very	1	0.5
Very much	1	0.5

**Table 2 jfmk-08-00091-t002:** Summary statistics for the score variable of the VAS scale of the three groups relating to the months of distance learning.

	1–6 Months	6–12 Months	>12 Months
VAS scale (mean ± SD)	3.67 ± 1.78	4.55 ± 1.62	4.28 ± 2.24

**Table 3 jfmk-08-00091-t003:** ANOVA model for distance learning and residuals.

	Df	Sum Sq	Means Sq	F Value	Pr (>F)
Distance learning	2	12.7	6.370	1.729	0.18
Residuals	198	729.5	3.648		

## Data Availability

Data are included in the manuscript.

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
