# Peer review of "The Onset of Musculoskeletal Pain in the COVID-19 Era: A Survey of Physiotherapy Students in Sicily"

_jfmk, 2023, doi:10.3390/jfmk8030091_

Round 1

Reviewer 1 Report

The manuscript is interesting and quite well written. I have few comments:

1- Abstract. Keywords: musculoskeletal pain; COVID-19 era; physiotherapy studies; study hours; questionnaires. Please, insert the abstract in the on-line first page

2- Results: L25-30. A total of 128 questionnaires were collected. More than half of  the students (51.6%) reported dedicating 15–22 h per week to distance learning for a duration of 6– 12 months (50%). Regarding study location, most students preferred studying at a desk (82.8%), with  slightly over half (57.8%) adopting a backrest while studying remotely. Analysis of the students’  posture during study hours revealed common positions, including tilting the head forward by more  than 20 degrees (47.8%), leaning the trunk forward by more than 20 degrees (71.9%), both shoulders  being hunched forward (57.0%), wrists positioned above the level of the elbows (46.1%), thighs  pointing upwards (41.4%), and one or both feet in a downward or dorsiflexed position (69.5%). Please add the most imporatnt statistically significant values to support the data.

3) Conclusion: L31-35. The questionnaire responses indicate that the lifestyle of university students, influenced  by online teaching, has deteriorated, leading to musculoskeletal pain, including myofascial pain. These results are primarily influenced by the adopted posture and the duration of time spent in  these positions. Please, improve this paragraph.

4) Introduction. L43-46. Coronavirus disease 2019 (COVID-19) emerged at the end of 2019, profoundly impacting the global population [1]. The widespread transmission of the virus is evident in its rapid spread, reaching nearly all countries within a span of fewer than 6 months [2]. I suggest to improve this paragraph with some information and discuss these references:

a-Minimal Clinically Important Differences in Inspiratory Muscle Function Variables after a Respiratory Muscle Training Programme in Individuals with Long-Term Post-COVID-19 Symptoms. J Clin Med. 2023 Apr 5;12(7):2720. doi: 10.3390/jcm12072720.

b- Different Methods to Improve the Monitoring of Noninvasive Respiratory Support of Patients with Severe Pneumonia/ARDS Due to COVID-19: An Update. J Clin Med. 2022 Mar 19;11(6):1704. doi: 10.3390/jcm11061704. 

c- Anthropometric Measurements and Admission Parameters as Predictors of Acute Respiratory Distress Syndrome in Hospitalized COVID-19 Patients. Biomedicines. 2023 Apr 18;11(4):1199. doi: 10.3390/biomedicines11041199. 

3) Introduction. L 144-145. This study aimed to examine the experiences of university students during this significant historical period, which will undoubtedly leave a lasting impression in the collective memory. Improve the description of study aim and underline the novelty of the study. 

4) 2.4 Statistical analysis L183-195. The raw data from Google Forms was extracted and subsequently exported to Excel for  statistical analysis. A descriptive analysis was performed, calculating frequencies and relative percentages of the various variables relevant to the survey. The aim was to assess whether there were differences among subjects based on the duration of their engagement in distance learning and the intensity of pain reported on the Visual Analog Scale (VAS). Specifically, a one-way analysis of variance (ANOVA) was utilized, which is a statistical  method suitable for testing differences between the means of three or more groups. In our case, the three groups were defined based on the duration of distance learning: 1–6 months, 6–12 months, and over 12 months. One of the assumptions underlying the ANOVA  method is the homoscedasticity or equal variance between groups. To test this hypothesis,  Leneve’s test was employed as an alternative to Bartlett’s test, as it is less sensitive to deviations from normality. All data were analyzed using R Software (Version 4.1.1). Please, add the statistically significant value of p.

5) Figure 2: Bar graph of Q.9: Study tools. Please, improve the explannation of this figure. 

6) Could you please improve the legend of all figures.

7) 4. Discussion L316-319. Almost all the participants in this study dedicated between 15 and 22 h a week to distance  learning for a period of 6 to 12 months. The requirement to attend online lessons necessitated spending extended periods in front of computers, tablets, and mobile phones, leading to a sedentary lifestyle that is considered inappropriate for their well-being. Please, summarise here the most important results of the study.

8) Please, insert and discuss the figures in the results.

9) L447-452. The study had several limitations, including only Sicilian university students who were  predominantly recruited through the “WhatsApp” application. The exact percentage of active participants on this social network remains unclear. Additionally, there is a possibility of recall bias and social desirability bias due to the self-reported and retrospective nature of the data. 

10) L 452 5. Conclusions. Could you please underline the novelty of the study and the possible clinical implications.

Author Response

Thank you for your valuable suggestions. I proceeded to perform the requested revisions that you will find underlined in yellow in the text I uploaded.

First part:

1)  I have inserted the abstract in the first online page.

2)  I have entered the most important statistically significant values

3)  I have improved the conclusions

4)  I have improved the paragraph with references

Second part:

3) Introduction:  The novelty of the study is to try to identify the correlations between postures taken during the pandemic and associated musculoskeletal disorders

4)  Significant p value has been defined.

5) -  6)  Figures: I improved the figures and legends

7)  I have summarised the relevant data

8) Figures have been inserted in the results

9)  As stated in the study we know the limitations of the study and the possible related biases

10)  I stressed the novelty of the study

Reviewer 2 Report

JFMK-2456652

Thank you for the opportunity to review this paper.

Manuscript Title:The onset of Musculoskeletal pain in the COVID-19 era: Survey of physiotherapy students in Sicily”.

Overview: The aim of this study was investigate the Effects of remote learning on Sicilian physiotherapy students during the COVID-19 pandemic, specifically focusing on the occurrence of Musculoskeletal pain.

General comments: This is an interesting manuscript that addresses an important area. Please see my specific comments below for more details.

Specific comments:

1.       Introduction: The introduction has too much information, it seems the introduction of a final degree project rather than a scientific article. Please restructure this section.

For example: Lines 110 to 124 it is the same reference… [18]… lines 125- 131 ref[19] Lines 132- 143 ref [4,5,9,12]… please, Summarize and restructure the entire section (1. Introduction).

  1. The keywords are absolutely fine.
  2. Materials and Methods: Regarding the material and methodology I have several questions for the authors:

-          The authors do not present an approval of the corresponding Ethics Committee for the realization of this study. Why?

-          Informed consent was signed? Information about informed consent must appear in the questionnaire itself, and the person who agrees to complete the questionnaire is giving consent. But it must appear.

-          Where can I find the questionnaire? From which validated questionnaires has it been created?

-          I like that the questionnaire has been sent to more than one university center so I get more sample.

-          A first questionnaire validated by experts was carried out before launching it to the students ...?  How have you validated your questionnaire?

  1. I consider important the Discussion section where authors reflect on the literature published so far on this topic.

The topic in question is interesting, perhaps already a little out of time, since we are in mid-2023, but I understand the work and time dedicated to research. The objective of publishing it now is to continue with the topic comparing perhaps with the state of students after Covid-19 ...?

Thank you

Author Response

Thank you for your valuable suggestions. I proceeded to perform the requested revisions that you will find underlined in green in the text I uploaded.

1) We have modified the introduction

2) - We did not present the approval of the Ethics Committee, because it is an observational study.

  • The person who agrees to complete the questionnaire does not sign informed consent because the answers are in anonymous form, in order to do not compromise the answers.
  • Attached you will find the questionnaire
  • Before submitting the questionnaire to the students, this was validated by experienced staff.
  • The statistical comparison with the status of students post-covid 19 will be provided at the next manuscript.

The discussion has been modified

Round 2

Reviewer 2 Report

Thank you for your reply.

But... Adding reference numbers in the same paragraph is NOT"modify the introduction", the authors present complete paragraphs with the same bibliographic reference:

For example, Introduction: 

Lines 55-60 ref [6]

Lines 60-63 ref [7]

Lines 64-70 ref [8]

Lines 75-81 ref [11]

Full paragraphs with the same reference. 

The authors confirm that the Discussion has been modified. I am sorry but Sorry, but add two figures (9 and 13) and four lines ... is not modifying a discussion.

Author Response

Thank you for your valuable suggestions. I apologize for not noticing your comment about the discussion section, in fact the revisions underlined in yellow were not related to your review but to the requests of a second reviewer. I proceeded to carry out the revisions as requested by both the discussion and the introductory.